# Differential Run-Length Encryption in Sensor Networks

**DOI:** 10.3390/s19143190

**Published:** 2019-07-19

**Authors:** Chiratheep Chianphatthanakit, Anuparp Boonsongsrikul, Somjet Suppharangsan

**Affiliations:** Department of Electrical Engineering, Faculty of Engineering, Burapha University Chonburi Campus, Chonburi 20131, Thailand

**Keywords:** data compression, wireless sensor networks, energy consumption

## Abstract

Energy is a main concern in the design and deployment of Wireless Sensor Networks because sensor nodes are constrained by limitations of battery, memory, and a processing unit. A number of techniques have been presented to solve this power problem. Among the proposed solutions, the data compression scheme is one that can be used to reduce the volume of data for transmission. This article presents a data compression algorithm called Differential Run Length Encryption (D-RLE) consisting of three steps. First, reading values are divided into groups by using a threshold of Chauvenet’s criterion. Second, each group is subdivided into subgroups whose consecutive member values are determined by a subtraction scheme under a K-RLE based threshold. Third, the member values are then encoded to binary based on our ad hoc scheme to compress the data. The experimental results show that the data rate savings by D-RLE can be up to 90% and energy usage can be saved more than 90% compared to data transmission without compression.

## 1. Introduction

Wireless sensor networks (WSNs) consist of smart wireless sensors working together to monitor areas and to collect data such as temperature and humidity from the environment. However, sensor nodes are faced with resource constraints in terms of energy, memory, and a processing unit [1]. Many works [2,3,4] proposed a variety of solutions to overcome the restrictions. A challenge is how to prolong sensor life time during data delivery from sensor nodes to a base station. Energy consumption is a hard problem in the design and deployment in WSNs because sensor nodes may be deployed in harsh environments where it is not easy to replace the batteries [5]. Energy consumption mostly occurs in either data computing or data transmission. Wang et al. [6] reported that the ratio between computing and communication incurred energy consumption is about 1:3000; therefore, sensor nodes should focus on effective data communication. If sensor nodes reduce the number of data transmissions, this can obviously save energy consumption in the entire network. The end-to-end energy cost and network lifetime are greatly restricted if the cooperative transmission model is not designed properly [7]. The most common technique for saving energy is the use of a sleep-wake scheduling scheme [8] in which a significant part of the sensor’s transceiver is switched off. However, the solution induces a problem of time synchronization [9] and the possibility of retransmitting data. Sensor network topologies also have a massive impact on energy usage in data transmission. In tree-based topologies [10], data aggregation approaches are often mentioned in order to reduce data redundancy and resulted in decreasing the number of data transmissions. However, it is merely given an approximate data value in a local area [11]. In cluster-based topologies [7,12], a cluster head plays an important role that collects and forwards all data from neighboring nodes to the base station. The cluster head consumes higher energy than other neighboring nodes and results in failures if it is out of energy more quickly. If the number of failures exceeds the tolerance level, a system may collapse [13]. To disregard this problem, a cluster head [14] is assumed as a special node having more sufficient energy than its neighboring nodes. On the other hand, our proposed solution does not require a special node. It simply can be applied for all sensor nodes including a cluster head that can be exhausted when it works hard. Residual energy is a criterion in selection of a cluster head [15].

In this paper, data compression is a proposed solution that can reduce a data packet size and amount of data transmission and result in prolonging the battery life of wireless sensor nodes. The proposed concept shown in Figure 1 can apply either lossless or lossy data compression. Furthermore, the proposed data compression does not require extra RAM. Data compression can be divided into lossless and lossy algorithms. Lossless compression provides data accuracy but normally requires extensive use of memory for making a lookup table. A sensor LZW (S-LZW) algorithm [16] is an extension of a lossless data compression algorithm created by Abraham Lempel, Jacob Ziv, and Terry Welch (LZW) [17,18]. Capo-Chichi et al. [19] and Roy et al. [20] reported a concept of S-LZW that is a dictionary-based algorithm that is initialized by all standard characters of 255 ASCII codes. However, a new string in the input stream creates a new entry and results in the limitation of memory in a sensor node. In [21,22,23,24], their schemes are based on lossless entropy compression (LEC) for data compression by using the Huffman variable length codes. The data difference is an input to an entropy encoder. LEC is one of the efficient schemes in data compression, therefore LEC is applied for reliable data transmission to monitor a structural health in wireless sensor networks [25]. On the other hand, lossy compression [26] is data compression that is appropriate for sending approximate data or repeated data [27]. The *K*-Run-Length Encoding (*K*-RLE) algorithm [28] is a lossy compression that is an adaptation of RLE [17]. *K*-RLE’s data accuracy and compression ratio depend on the K-precision.

## 2. Related Work

In [26], researchers presented a comparison of data compression schemes with different sensor data types and sensor data sets in WSNs. In [24], data types in compression can be considered and divided into smooth temperature and relative humidity data and dynamic volcanic data that exhibit dramatic different characteristics. Since power consumption is one of the main concerns, Koc et al. [29] studied and measured power consumption during data compression by using the MSP432 family of microcontrollers. With the same fixed parameters of wireless environments, the energy usage for a fixed size packet would be the same on delivering the packet. Reducing the number of data packets would help reduce the energy consumption. Therefore, data compression has played a significant role in WSNs. Two types of data compression are generally categorized and referred to as lossy and lossless compression. The former compression permanently removes a certain amount of data, reducing the size of data to much smaller than the original ones, but it degrades the quality of data. While the latter compression reduces the data size without any data quality loss, its compression rate is lower than the former compression. In our experiments, we compared our results with three lossless algorithms that is LEC [21], Lempel-Ziv-welch (LZW) [17,18] and run-length encoding (RLE), whereas we compared our results with *K*-RLE for the lossy scheme. LEC is a lossless algorithm based on the Huffman concept in which the entropy is used for defining the Huffman codes. Table 1 shows the prefix and suffix codes used in LEC. LEC computes the difference data values and then replaces the difference by the corresponding codes from Table 1. LEC Algorithm is shown in Algorithm 1.

**Algorithm 1** LEC Pseudocode**Require:**di, Table of LEC codes**Ensure:**bsi  **if** (di==0) **then**    ni=0  **else**    ni=⎡log2(|di|)⎤+1  **end if**  si = Table(ni) ▹ extract si from LEC Table  **if** (ni==0) **then**    bsi=si    **return**
bsi  **end if**  **if** (di>0) **then**    ai=(di)|ni▹(v)|ni is the ni low-order bits of *v*.  **else**    ai=(di−1)|ni  **end if**  bsi=(si,ai)  **return**
bsi

When $d_{i}$ is negative, low-order bits of the two’s complement representation of (d−1) are used for the suffix code. For example, suppose we have a data set: <19,18,20,21>. Starting with the first data 19, we then compute the value difference between a pair of consecutive data, resulting in <19,−1,2,1>. By using Table 1 and Algorithm 1 above, we obtain the following encoded $bs_{i}$: (0001 0011), (010,0) (011,10) (010,1).

LZW is a dictionary-based lossless compression. LZW used in the experiments begins with the value of 256 onwards to avoid repeating the value of the first 256 ASCII codes. The algorithm repeatedly reads a symbol input to form a string and checks if the string is not in the dictionary. Once such a string is found, the corresponding output code for the string without the last symbol that is the longest string in the dictionary is sent out, and the new found string is added to the dictionary with the next available output code. Table 2 shows an example of data input string AAAABAAAABCC. Applying Algorithm 2, the seven new codes (code from 256 to 262) are added into the dictionary and the output strings are <A, AA, A, B, AAA, B, C, C>. The output codes for those output strings are <65, 256, 65, 66, 257, 66, 67, 67> where each output code uses nine bits, so in total the encoding output uses 72 bits compared to original input 96 bits. Nevertheless, a larger dictionary requires larger memory.

**Algorithm 2** LZW Pseudocode  initialize Dictionary[0-255] = first 256 ASCII codes  STRING ← get input symbol  **while** there are still input symbols **do**    SYMBOL ← get input symbol    **if** (STRING+SYMBOL is in Dictonary) **then**      STRING = STRING+SYMBOL    **else**      output the code for STRING      add STRING+SYMBOL to Dictionary      STRING = SYMBOL    **end if**  **end while**  output the code for STRING

RLE is the simplest compression, working by counting the amount of repeating consecutive identical data. The amount of consecutive identical data followed by the data symbol is replaced for the original repeating data. For example, the data of AAABBCEEFFFFFFFFAA are compressed to 3A2B1C2E8F2A, which implies that there are 3 A’s, 2 B’s, C, 2 E’s, 8 F’s and 2 A’s next to each other in series. RLE pseudocode is shown in Algorithm 3.

**Algorithm 3** RLE Pseudocode  **while** there are still input symbols **do**    count=0    **repeat**      get input symbol      count=count+1    **until** symbol unequal to next symbol    output count and symbol  **end while**

*K*-RLE is based on a RLE algorithm allowing quality loss to such an extent. The value of *K* indicates the range of different data values. If *K*=1, for example, the data of <19,18,20,21> will be encoded as <(3,19),(1,21)> because the first three pieces of data are in the range of 19±1 and the last two pieces of data are in the range of 20±1. The encoded data <(3,19),(1,21)> then will be decoded as <19,19,19,21>. Obviously, the decoded data are different from the original data due to the lossy scheme. *K*-RLE pseudocode is shown in Algorithm 4.

**Algorithm 4***K*-RLE Pseudocode  v1← read input value  count=1  **while** there are still input values **do**    v2← read next input value    **if** (|v1−v2|≤K) **then**      count=count+1    **else**      output (count,v1)      v1←v2      count=1    **end if**  **end while**  output (count,v1)

LZW compresses the same data with the same encoding though these data are at different positions in the data input stream. In contrast, RLE requires that the same data must stand next to each other in a row. Both LEC and LZW apply a similar concept in terms of the prefix codes. However, LEC also has suffix codes addressing the different value; hence, each encoding of difference value in the input stream consists of prefix and suffix codes. If the input stream has many pieces of consecutive identical data, RLE performs very well. LZW would be preferred to RLE if the input stream consisted of many repeating data with shorter output codes. LEC works even better if the input stream has many of the same levels of the difference values with shorter prefix and suffix codes. Aforementioned algorithms have the linear time complexity O(n) and perform best in their own characteristics, not for general datasets. This gap stimulates how we can combine each strong point of those algorithms to compress the data. To this end, we have developed Differential Run Length Encryption (D-RLE), which also has the linear time complexity O(n), and will explain its concept in the next section.

## 3. Differential Run Length Encryption

This section presents the proposed algorithm called Differential Run Length Encryption or D-RLE, which consists of three steps. First, raw data are collected and divided into several groups by using a threshold of Chauvenet’s criterion. Second, consecutive data are subtracted and arranged into multiple subgroups of differential values based on a threshold of *K*-RLE. Third, our adaptive data compression, which is the DSC-based scheme [30], is employed to each piece of subgroup data. Data formats in D-RLE are shown in Table 3. As a result, D-RLE significantly reduces the amount of data delivery, saves energy consumption and prolongs the battery life of the sensor nodes.

### 3.1. Group Division by Chauvenet’s Criterion

The raw data can be considered as a random sample <$r_{0},r_{1},\ldots ,r_{m}$> and they are grouped by using a pre-defined Chauvenet’s criterion [31,32]$D_{max}$, which is set to 1.96 according to the significance level of 0.05 from statistics. The data $r_{i}$ is passed to Equation (Equation 1) for calculating $D_{i}$ in which μ and σ are mean and standard deviation, respectively. The value of $D_{i}$ is then compared with $D_{max}$ to consider if $r_{i}$ should belong to the same group or not. We maintain $r_{i}$ to the same group if $D_{max}$ is greater than or equal to $D_{i}$; otherwise, we split $r_{i}$ into the next group. For example, suppose we have the following raw data <21,25,28,30,31,35,37,42,47,49,50,55,62,76,82,95,105,103,92,86,71,63,59,52,41,34,30,26,25,21>. As the raw data coming into Equation (Equation 1), we found that $D_{15}=\left|95-52.43\right|/25.34=1.68,$ whereas $D_{16}=\left|105-52.43\right|/25.34=2.07.$ Since $D_{max}=1.96$ is less than $D_{16}=2.07,$ data $r_{16}$ is split into a different group. Therefore, in this data sample set, there are two groups where the first group is <21,25,28,30,31,35,37,42,47,49,50,55,62,76,82,95>, and the other group is <105,103,92,86,71,63,59,52,41,34,30,26,25,21>.
(1)$$D_{i}=\frac{|r_{i}-\mu |}{\sigma }$$

Once a group is split, the mean and standard deviation are recalculated and updated with the remaining data and used in Equation (Equation 1) for the next round of group divisions. Algorithm 5 shows the algorithm for the group division step.

**Algorithm 5** Group Division**Require:**r[m+1]={r0,r1,…,rm}**Ensure:**G={G1,G2,⋯,Gg}  initialize N=m+1,Dmax=1.96,g=1  initialize arrays Di[N]  compute μ and σ  add r0 into G1  **for** (i=1tom) **do**    Di[i]=|ri−μ|σ    **if** (Dmax<Di[i]) **then**      g=g+1      add ri into Gg      update μ,σ    **else**      add ri into Gg    **end if**  **end for**

### 3.2. Subgroups Division

Each group from the first step will be sub-divided based on the *K*-RLE scheme. The value of *K* implies the degree of tolerance. It is lossless if *K* equals zero. The result of the subgroup division is written in the compact format: <$r_{b},|c_{1},d_{1}{|}_{1},|c_{2},d_{2}{|}_{2},\ldots ,{\left|c_{n},d_{n}\right|}_{n}$>, where the symbol $|c_{i},d_{i}{|}_{i}$ is used to separate *i*th subgroup and $r_{b}$ is the first value of the group. The number of elements in *i*th subgroup is denoted by $c_{i}$ and the value difference between the last raw data of the present subgroup *i*th and the last raw data in the previous subgroup (i−1)th is denoted by $d_{i}$. The algorithm of discovering the $c_{i}$ and $d_{i}$ is shown in Algorithm 6. As we start off the index from zero, m+1 is the number of data members in the group. The index *n* is the number of subgroups that is also the number of members in set $C_{i}$ and $D_{i}$. The relationship between *m* and $c_{i}$ is shown by Equation (Equation 2):(2)$$m=\sum_{i=1}^{n}c_{i}.$$

**Algorithm 6** Computing |ci,di|**Require:**K,r={r0,r1,…,rm}**Ensure:**Ci={c1,c2,…,cn},Di={d1,d2,…,dn}  initialize i=1,j=1,count=1,f=1,s=0  **while**j<(m+1)**do**    s=r[f]−r[j+1]    **if**|s|≤K**then**      count=count+1    **else**      C[i]=count      D[i]=r[j]−r[f−1]      i=i+1      f=j+1      count=1    **end if**    j=j+1  **end while**

For the first group <21,25,28,30,31,35,37,42,47,49,50,55,62,76,82,95>, we ran the algorithm from Algorithm 6 above and had the following subgroup division result: <21,|1,4|,|1,3|,|2,3|,|1,4|,|1,2|,|1,5|,|1,5|,|2,3|,|1,5|,|1,7|,|1,14|,|1,6|,|1,13|>. Value 21 is the first raw data $r_{b}$ of the group, followed by $|c_{i},d_{i}{|}_{i=1to13}$. The variable $c_{i}$ is just a counter, starting from one, indicating how many pairs that the absolute difference between the first and next raw data in the same subgroup do not differ more than the pre-defined *K* value. The variable $d_{i}$ roughly dictates to us how different the data are between the present and previous subgroups. The following subgroup division result: <105,|1,−2|,|1,−11|,|1,−6|,|1,−15|,|1,−8|,|1,−4|,|1,−7|,|1,−11|,|1,−7|,|1,−4|,|2,−5|, |1,−4|> is obtained for the second group from the first step.

### 3.3. Adaptive Data Encoding

The last step is the process of adaptively encoding each subgroup division result. Shortened opcodes for $c_{i}$ and $d_{i}$ are used to compress raw data via the encoding process. The number of bits to represent $c_{i}$ and $d_{i}$ is determined by set $C_{i}=\left\{c_{1},c_{2},\ldots ,c_{n}\right\}$ and set $D_{i}=\left\{d_{1},d_{2},\ldots ,d_{n}\right\}$, respectively. We shall use the first subgroup division result, <21,|1,4|,|1,3|,|2,3|,|1,4|,|1,2|,|1,5|,|1,5|,|2,3|,|1,5|,|1,7|,|1,14|,|1,6|,|1,13|>, as an illustration of the encoding process. To begin with, set $C_{i}$ is {1,1,2,1,1,1,1,2,1,1,1,1,1} and set $D_{i}$ is {4,3,3,4,2,5,5,3,5,7,14,6,13}. Then, Equations (Equation 3)–(Equation 5) are used to compute the number of bits for $c_{i}$ and $d_{i}$, respectively: (3)$$\begin{array}{cc} \#c & =⎡\log_{2}\left(\max_{i=1}^{n}\left(c_{i}\right)\right)⎤, \end{array}$$
(4)$$\begin{array}{cc} d_{L} & =d_{i}\;\mathrm{where}\;i=\underset{i}{\mathrm{argmax}}\left|D_{i}\right|, \end{array}$$
(5)$$\begin{array}{cc} \#d & =\left\{\begin{array}{cc} k+1 & \mathrm{if}\;-\left(2^{k}\right)+1\leq d_{L}\leq 2^{k},\\ k+2 & \mathrm{if}\;d_{L}=-\left(2^{k}\right), \end{array}\right.\;\mathrm{where}\;k\in I^{+}. \end{array}$$

Because the maximum value of C is 2, the number of bits for $c_{i}$ equals $⎡\left(\log_{2}2\right)⎤$ = 1 bit. The argument of the maximum for index *i* of absolute value of $D_{i}$ is i=11; hence, $d_{L}=d_{11}=14$ and $14\leq 2^{k=4}$; then, the number of bits for $d_{i}$ equals k+1=4+1=5 bits. In binary code, 1 and 5 are represented by 0001 and 0101, respectively. We combine 0001 with 0101 to form a byte as 0001 0101 referred to as <|#c,#d|> and this byte will be our length field to notify the decoder of the bit sizes of $c_{i}$ and $d_{i}$. We use these bit sizes to limit the bit length used for binary codes of $c_{i}$ and $d_{i}$. In addition, the value of each $c_{i}$ before changing to binary code is decreased by one as doing so will help reduce bit sizes for $c_{i}$. For example, the binary code $c_{i}=8$ is 1000 (4 bits), but we could obtain the shortened binary code 111 (3 bits) if we decrease 8 to 7. We could not do the same decrease for $d_{i}$ as $d_{i}$ can be either positive or negative values, whereas $c_{i}$ is only positive integers. Subsequently, the data field is set to the format <$r_{b},|c_{1},d_{1}{|}_{1},|c_{2},d_{2}{|}_{2},\ldots ,{\left|c_{n},d_{n}\right|}_{n}$> in which $r_{b}$ is represented by its corresponding 8-bit binary code, and $|c_{i},d_{i}{|}_{i=1\;\mathrm{to}\;n}$ are opcodes for $c_{i}$ and $d_{i}$ denoted by shorten binary codes where $c_{i}=c_{i}-1$. Two’s complement is used for negative values of $d_{i}$. The length and data fields join together to make a data payload. Finally, the data payload <$|\#c,\#d|,r_{b},|c_{1},d_{1}{|}_{1},|c_{2},d_{2}{|}_{2},\ldots ,{\left|c_{n},d_{n}\right|}_{n}$> of our example is <${\left|1,5|,21,|0,4\right|}_{1}{,|0,3|}_{2}{,|1,3|}_{3}{,|0,4|}_{4}{,|0,2|}_{5}{,|0,5|}_{6}{,|0,5|}_{7}{,|1,3|}_{8}{,|0,5|}_{9}{,|0,7|}_{10}{,|0,14|}_{11}{,|0,6|}_{12},{\left|0,13\right|}_{13}$> and will be encoded as <${\left|0001,0101|,|0001\;0101|,|0,00100\right|}_{1}{,|0,00011|}_{2}{,|1,00011|}_{3}{,|0,00100|}_{4},{\left|0,00010\right|}_{5}$,${\left|0,00101\right|}_{6}{,|0,00101|}_{7}{,|1,00011|}_{8}{,|0,00101|}_{9}{,|0,00111|}_{10}{,|0,01110|}_{11}{,|0,00110|}_{12},{\left|0,001101\right|}_{13}$>. The total number of encoded bits for each subgroup division result can be found by Equation (Equation 6). In our example case K=1, the total number of encoded bits is 16+[13∗(1+5)]=94 bits compared to 2∗16∗8=256 bits without compression. This means we have saved approximately 63.28% of data delivery:(6)$$Size_{bit}=16+\left[n\times \left(\#c+\#d\right)\right].$$

For the second subgroup division result, the number of bits for $c_{i}$ and $d_{i}$ equal to 1 and 5 bits, respectively. Therefore, the data payload of the second subgroup is <|0,5|,105,|0,−2|,|0,−11|,|0,−6|,|0,−15|,|0,−8|,|0,−4|,|0,−7|,|0,−11|,|0,−7|,|0,−4|,|1,−5|,|0,−4|> and encoded as <${\left|0000,0101|,|0110\;1001|,|0,11110\right|}_{1}{,|0,10101|}_{2}{,|0,11010|}_{3}{,|0,10001|}_{4}{,|0,11000|}_{5}{,|0,11100|}_{6}{,|0,11001|}_{7}{,|0,10101|}_{8}{,|0,11001|}_{9}{,|0,11100|}_{10}{,|1,11011|}_{11},{\left|0,11100\right|}_{12}$>. The total number of encoded bits is 16+[12∗(1+5)]=88 bits compared to 240 bits without compression for the second subgroup. This means we have saved approximately 63.33% of data delivery.

This example shows that, from the raw data size of 256+240=496 bits, our D-RLE has compressed the raw data to 94+88=182 bits, which means we have saved bits sent up to 63.31% of the raw data. Note that we have demonstrated only 30 pieces of raw data for illustration purposes, but, in practice, each sensor node would collect more data in the long run before delivering the data. In the next section, we will show that D-RLE impressively improves accomplishment for the longer data delivery.

## 4. Performance Evaluation

In our testbed, we assumed that data sets are already stored in a sensor node. The sensor node could be a cluster head that collects data either from itself or from neighboring nodes. Later, the sensor node performs data compression and wirelessly sends compressed data to a receiver. This section analyzes the performance of our algorithm and compares the performance with four benchmark algorithms on five datasets.

### 4.1. Effect of *K* Value

We have evaluated the performance of our algorithm in terms of data rate savings (DRS) [30] as shown in Equation (Equation 7). Particularly, the case of lossless compression (K=0) and the case of lossy compression (K=1) were evaluated. The lossy case for K=1 has been shown in the previous section and its DRS is 63.31%. We did repeat the same procedure except for K=0 in the lossless case on the same raw data <21,25,28,30,31,35,37,42,47,49,50,55,62,76,82,95,105,103,92,86,71,63,59,52,41,34,30,26,25,21>, and received the corresponding encoded data payload <$|0,5|,r_{b},d_{1},d_{2},\ldots ,d_{n}$>. Notice that the number of bits for the opcode $c_{i}$ is 0; hence, there is no need for sending $c_{i}$ of each subgroup. The total number of encoded bits used is 172. Therefore, the DRS of the lossless case is 172/496=65.32%, compared to 63.31% of the above lossy case:(7)$$DRS=\left(1-\frac{\mathrm{size}\;\mathrm{of}\;\mathrm{compressed}\;\mathrm{data}}{\mathrm{size}\;\mathrm{of}\;\mathrm{raw}\;\mathrm{data}}\right)\times 100\%.$$

To achieve more data rate savings, we could allow more data lost or distorted to some extent. It depends on applications how much the quality loss is acceptable and this is done via the value of *K* in the algorithm. Table 4 shows the effect of varying *K*. We obtained more data rate savings when K≥4 for the same raw data previously illustrated.

### 4.2. Evaluation Results

We have split our experiments into two sets. The dataset we used in the first set is varied in size—roughly speaking as small, medium, and large sizes—while, for the second set, we have fixed each dataset with the same size. Subsequently, benchmark algorithms as well as our proposed D-RLE algorithm were applied to compress those datasets and then the energy consumption for data compression and transmission would be measured.

Starting with the first set, we have simulated a 100-byte temperature dataset shown in Figure 2c and its shape resembles Figure 2d, which is a real collected dataset by [33]. In addition to these datasets, three more datasets referred to as sine-like, chaotic, and temperatureMin datasets as illustrated in Figure 2 were used for evaluating effectiveness of compression algorithms in our experiments. The simulated temperature dataset was created by Algorithm 7 while the temperatureHr dataset was the actual hourly recorded temperature data for 48 h. The temperatureMin dataset was retrieved from the same source of the temperatureHr dataset, but minutely recorded data. For the sine-like dataset, the minimum and maximum data values are 2 and 19, respectively. The neighboring data value next to 2 is 3 and then the data value is increased by 3 until reaching the maximum value. The data value after the maximum was set to 18 then decreased by 3 until touching the minimum value. By doing so for two cycles, the sine-like dataset has 30 pieces of data. We have fixed data ranging from 2 to 19 for the sine-like dataset, whereas we have randomly selected data ranging from 0 to 20 for the chaotic dataset. For chaotic and simulated temperature datasets, each dataset consists of 100 pieces of raw data. The temperatureHr dataset only has 48 pieces of data as the data were hourly recorded for 48 h, whereas the temperatureMin dataset has 2880 pieces of data since the data were minutely recorded for the same 48 h period. The raw data in each dataset were recorded as a series of strings; hence, bit sizes of data 2, 12, and 102 are 8, 16, and 24 bits, respectively.

**Algorithm 7** Creating simulated temperature data**Require:**g,uppertemp,lowertemp**Ensure:**t[g]={t1,t2,…,tg}  initialize min=0,max=10,temp=20,i=1,up=true  **while**i≤g**do**    **while** (temp≤uppertemp)AND(upistrue) **do**      temp=temp+Random(min,max)      **if** (uppertemp<temp) **then**        up=false        temp=uppertemp      **end if**      t[i]=temp      i=i+1    **end while**    **while** (lowertemp≤temp)AND(upisfalse) **do**      temp=temp−Random(min,max)      **if** (temp<lowertemp) **then**        up=true        temp=lowertemp      **end if**      t[i]=temp      i=i+1    **end while**  **end while**

For the second set of experiments, we extended the size of each dataset except for temperatureMin dataset to 46,080 bits. The reason to do this experiment is to investigate the energy usage for lengthy data transmission in one shot compared to multishot transmission of a small amount of data. We expected that the one shot delivery should have more efficient energy usage than the multishot since the sensor node in the single shot would be in a silent or power saving mode longer, whereas the multishot could wake up the sensor node more frequently. For both sets of experiments, four selected benchmark compression algorithms which are RLE, *K*-RLE, LEC, and LZW were used and compared to our D-RLE algorithm. For *K*-RLE, we set K=1. A sensor node has been used in the experiments and run these algorithms to compress the datasets before transmitting data to a base station. The DRS obtained and energy consumed by each algorithm then were recorded for comparisons. The algorithms were implemented into a LAUNCHXL-CC1310 board [34] acting as the sensor node equipped with an RF module. To measure the power and energy usage in data compression and transmission, an MSP430FR5969 board [35] and code composer studio (CCS) program (version 8.0.0, Texas Instrument Inc., Dallas, Texas, USA) [36] were used. The MSP430FR5969 board was connected to the LAUNCHXL-CC1310 board as shown in Figure 3, and the amount of energy used was then measured by the EnergyTrace() function in CCS software. We took off the jumper connecting between the microcontroller and the debug parts of the CC1310 board to ensure that the power source came from the MSP430FR5969 board. The corresponding 3.3V Vcc and ground pins between the two boards were wired up as illustrated by black and white lines in Figure 3.

Table 5 and Table 6 show the comparison results and total energy consumption on the datasets with different sizes while Table 7 and Table 8 show the comparison results and total energy consumption on lengthy datasets with the same size of 46,080 bits, respectively. In the matter of DRS, the sine-like data gradually change values and there are no repeating values in adjacent data, so it is obvious that RLE and *K*-RLE poorly perform while D-RLE works much better than others. On the other hand, the temperatureMin dataset has many repeating values and this characteristic does help RLE, *K*-RLE, and D-RLE to have higher DRS than LEC and LZW. The temperatureMin dataset has many nearby identical repeating data making it possible for LZW to create a dictionary with shorter encoding bits than LEC, and hence LZW has compressed data better than LEC. For a simulated temperature dataset, LZW, RLE and *K*-RLE perform worse than LEC and D-RLE since there are no repeating data values. LEC and D-RLE share a similar concept in the way of encoding the difference values. While LEC considers the all of the data as one group for data encoding, D-RLE divides the data into several subgroups in which the members within the same subgroup are not much different, leading to smaller encoding bits. For the temperatureHr dataset, D-RLE performs very well while among *K*-RLE, LEC and LZW work comparably, but RLE is the worst. RLE is the worst algorithm for the datasets that there are no repeating values. The negative DRS of RLE means RLE could not compress data at all, and it also adds extra overheads into the raw data. For the chaotic dataset, LEC, LZW and D-RLE perform better than the RLE family due to the data fluctuation and the dataset rarely has repeating data. We found that, in terms of DRS on those datasets, our D-RLE is the winner on both single shot and multi-shot patterns. Though the compression time by D-RLE is longer than others except for LZW, the compression energy used by D-RLE is not much different from others in multi-shot patterns. For the single shot pattern, D-RLE spends compression energy similar to LEC and consumes more compression energy than RLE and *K*-RLE, but less than LZW. We would suggest using D-RLE for a dataset that has a long sequence of data and it works best for repeating data or a gradual change in data values.

As a result of highest DRS performance, the number of packets for data delivery by D-RLE is smaller than the number of packets by other algorithms, leading to less transmission energy. In terms of energy use, D-RLE uses the least total energy compared to other algorithms on most of the datasets as shown in Table 6 and Table 8. For example, in temperatureHr dataset of 46,080 bits, D-RLE approximately sends only 12 packets with the total energy use of 18.82 mJ, while others use more than 30 packets with the total of energy greater than 45 mJ. The total power use, $P_{total}$, consists of two parts from compression and transmission steps, which is referred to as compression power, $P_{c}$, and transmission power, $P_{t}$, respectively. $P_{total}$ is determined by Equation (Equation 8) in which the subscript *i* indicates the *i*th group number when we compress the data, and *j* expresses the *j*th payload or packet number that we deliver. The values of *g* and *p* are the number of groups and the number of packets. To calculate corresponding energy used in each step, we use the relationships between energy and power from Equations (Equation 9) and (10), where $T_{c_{i}}$ and $T_{t_{j}}$ are the compression time spending for compressing *i*th group and transmission time spending for delivering *j*th packet, respectively. Lastly, total energy consumption, $E_{total}$, is computed by Equation (11), adding transmission and compression energy. The experimental results show that D-RLE takes minimal transmission energy in exchange for slightly more compression energy, but it is worthy of being considered, as D-RLE significantly reduces total energy use while other benchmark algorithms consume much higher total energy level on the same datasets:
(8)$$\begin{array}{cc} P_{total} & =\sum_{i=1}^{g}P_{c_{i}}+\sum_{j=1}^{p}P_{t_{j}}, \end{array}$$
(9)$$\begin{array}{cc} E_{c} & =\sum_{i=1}^{g}P_{c_{i}}\times \Delta T_{c_{i}}, \end{array}$$
(10)$$\begin{array}{cc} E_{t} & =\sum_{j=1}^{p}P_{t_{j}}\times \Delta T_{t_{j}}, \end{array}$$
(11)$$\begin{array}{cc} E_{total} & =E_{t}+E_{c}. \end{array}$$

Figure 4 compares power and energy consumed by D-RLE during data compression and transmission between single shot and multishot cases on the TemperatureMin dataset, respectively. Both cases have the same data length of 46,080 bits. The single shot receives all of the data before starting compression and transmission while the multishot would receive several data portions in which each portion has the same data length. According to the graph, it is clearly seen that the transmission period demands higher power consumption than the compression period. However, the energy consumption during the transmission indicated as T in the graph is less than the energy used during compression indicated as C in the graph. The smaller size of compressed data gives the shorter period of data transmission time for the single shot case. On the other hand, the multishot case has to repeat many compression and transmission cycles and take more time. In each cycle of the multishot, the compression step is reinitiated, which gives us lower compression efficiency and hence the accumulated energy used by the multishot is greater than the accumulated energy used by the single shot. Therefore, the single shot is more efficient and energy saving compared to the multishot. Other datasets have the graphs in the same manner as the Temperature dataset.

### 4.3. Performance Visualization

We have plotted radar charts as shown in Figure 5 according to five categories for making a simple way to visualize performance comparison among the algorithms. The first three categories are DRS and data accuracy (in the sense of how much difference there is between the decoded data and its original data). The last three ad hoc categories are called compression time efficiency (CTE), compression energy efficiency (CEE), and transmission energy efficiency (TEE) in which they are defined by Equations (12)–(14), respectively. The parameter *A* in those equations is the number of algorithms we used in the experiments, i.e., A=5. $T_{c_{i}},E_{c_{i}}$ and $E_{t_{i}}$ are the compression time, compression energy, and transmission energy of *i*thalgorithm, respectively. Each category has a score from 0 to 100; the higher the score, the better the performance. The left panel of Figure 5 shows comparisons among RLE, *K*-RLE and D-RLE while the right panel of Figure 5 shows comparison among LEC, LZW and D-RLE. For the left panel, D-RLE performs better than the others on most categories except for CTE given the fact that D-RLE takes more time in the compression step. For the right panel, D-RLE performs equally or slightly better than the others in terms of CEE, TEE, and accuracy. D-RLE is located between LEC and LZW on CTE, whereas D-RLE mostly achieves better DRS (DRS results on the radar charts Figure 5a–c might not be clearly seen as shown as the number in Table 7). On average, D-RLE gets a high score and is well balanced, reaching the vertex of pentagon in the graph when compared to other algorithms in each category.
(12)$$\begin{array}{cc} CTE_{i} & =(1-\frac{T_{c_{i}}}{\sum_{a=1}^{A}T_{c_{a}}})\times 100 \end{array}$$
(13)$$\begin{array}{cc} CEE_{i} & =(1-\frac{E_{c_{i}}}{\sum_{a=1}^{A}E_{c_{a}}})\times 100 \end{array}$$
(14)$$\begin{array}{cc} TEE_{i} & =(1-\frac{E_{t_{i}}}{\sum_{a=1}^{A}E_{t_{a}}})\times 100 \end{array}$$

## 5. Conclusions

We have presented a compression algorithm called D-RLE applied to the domain of wireless sensor nodes in which energy use is one of the most important aspects. It starts with dividing the data into many groups based on Chauvenet’s criterion and then each group further forms subgroups to which an adaptive encoding is applied. According to the experimental results, D-RLE have demonstrated that it performs very well, gives the highest data savings rate and spends less energy compared to other benchmark algorithms. In particular, D-RLE is suitable for big amounts of data with repeating or gradually changed values and for a single shot delivery mode. Due to its highest compression rate, the amount of data transmission is significantly reduced and hence less energy is demanded. This prolongs the battery life of the sensor nodes. This work is an alternative way to increase the performance of the sensor node concerning the energy.

## Figures and Tables

**Figure 1 sensors-19-03190-f001:**
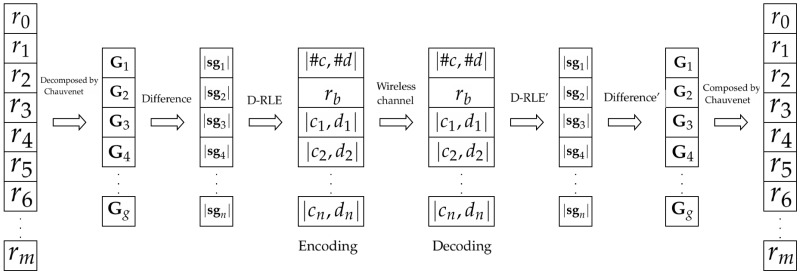
Data encoding and decoding process.

**Figure 2 sensors-19-03190-f002:**
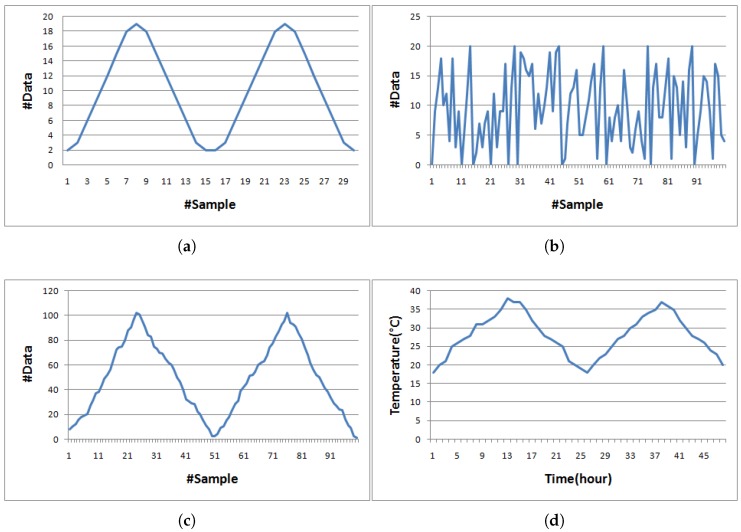
Five datasets used in the experiment. (**a**) sine-like; (**b**) chaotic; (**c**) simulated temperature; (**d**) temperatureHr; (**e**) temperatureMin.

**Figure 3 sensors-19-03190-f003:**
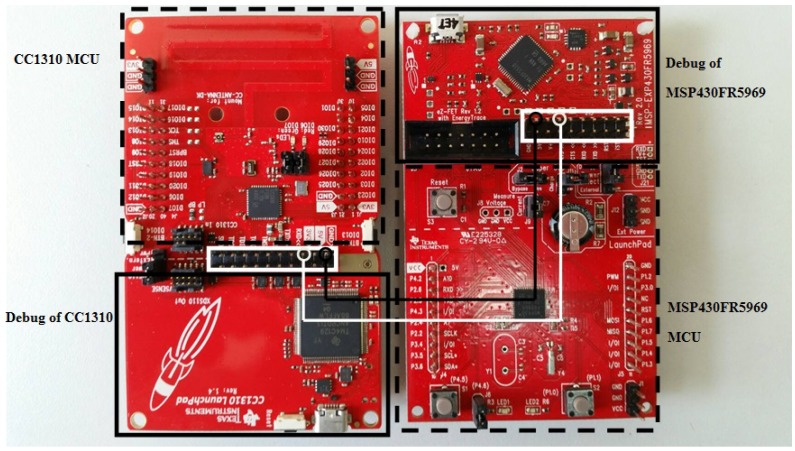
Board configuration for measuring energy use.

**Figure 4 sensors-19-03190-f004:**
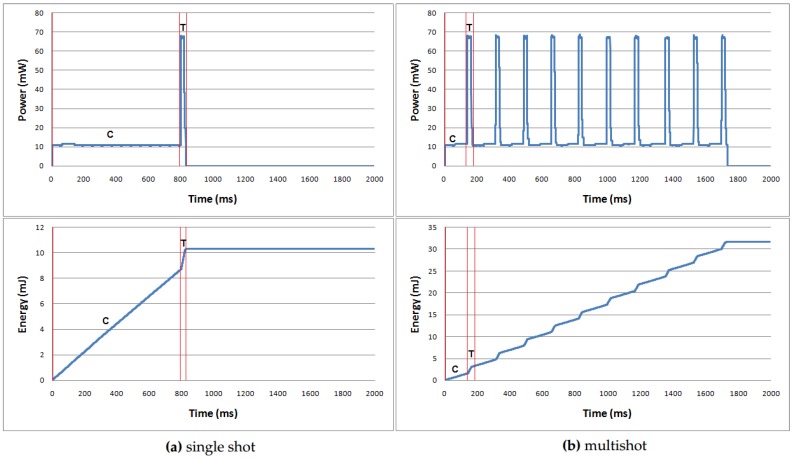
Power and energy comparison between single shot and multishot on the TemperatureMin dataset.

**Figure 5 sensors-19-03190-f005:**
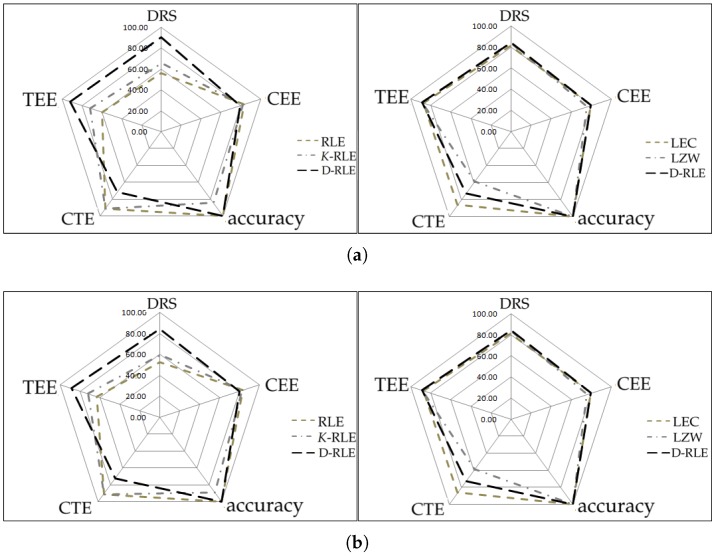
Radar chart comparison. (**a**) sine-like; (**b**) chaotic; (**c**) simulated temperature; (**d**) temperatureHr; (**e**) temperatureMin.

**Table 1 sensors-19-03190-t001:** LEC codes.

Level ($n_{i})$	Bits	Prefix ($s_{i}$)	Suffix Range ($a_{i}$)	Value ($d_{i}$)
0	2	00	-	0
1	4	010	0…1	−1, 1
2	5	011	00…11	−3, −2, 2, 3
3	6	100	000…111	−7, …, −4, 4, …, 7
4	7	101	0000…1111	−15, …,−8, 8, …, 15
5	8	110	00000…11111	−31, …, −16, 16, …, 31
6	10	1110	000000…111111	−63, …, −32, 32, …, 63
7	12	11110	0000000…1111111	−127, …, −64, 64, …, 127

**Table 2 sensors-19-03190-t002:** LZW codes.

String	Output	Dictionary	Total Bits
A	65	256 = AA	9
AA	256	257 = AAA	18
A	65	258 = AB	27
B	66	259 = BA	36
AAA	257	260 = AAAB	45
B	66	261 = BC	54
C	67	262 = CC	63
C	67		72

**Table 3 sensors-19-03190-t003:** Data formats in D-RLE.

Process	Format
raw data	$<r_{0},r_{1},\ldots ,r_{m}>$
group division	$<G_{1},G_{2},\ldots ,G_{g}>,$
	$\mathrm{where}\;G_{1}=<r_{0},r_{1},\ldots ,r_{j-1}>,G_{2}=<r_{j},r_{j+1},\ldots ,r_{k-1}>,\ldots ,G_{g}=<r_{l},r_{l+1},\ldots ,r_{m}>$
subgroup division	for each $G_{i}=g_{be}=<r_{b},r_{b+1},r_{b+2},\ldots ,r_{e}>$
	$\Rightarrow \;<r_{b},|{\mathrm{sg}}_{1}\left|,\right|{\mathrm{sg}}_{2}\left|,\ldots ,\right|{\mathrm{sg}}_{n}|>$
	$\Rightarrow \;<r_{b},|c_{1},r_{b+1},\ldots ,r_{b+c_{1}}\left|,\right|c_{2},r_{b+c_{1}+1},\ldots ,r_{b+c_{1}+c_{2}}\left|,\ldots ,\right|c_{n},r_{b+c_{1}+c_{2}+\ldots +c_{n-1}+1},\ldots ,r_{e}|>$
	$\Rightarrow \;<r_{b},|c_{1},\left(r_{b+c_{1}}-r_{b}\right)\left|,\right|c_{2},\left(r_{b+c_{1}+c_{2}}-r_{b+c_{1}}\right)\left|,\ldots ,\right|c_{n},\left(r_{e}-r_{b+c_{1}+c_{2}+\ldots +c_{n-1}}\right)|>$
	$\Rightarrow \;<r_{b},|c_{1},d_{1}\left|,\right|c_{2},d_{2}\left|,\ldots ,\right|c_{n},d_{n}|>$
encoded data	$<|\#c,\#d|,r_{b},|c_{1},d_{1}\left|,\right|c_{2},d_{2}\left|,\ldots ,\right|c_{n},d_{n}|>$

**Table 4 sensors-19-03190-t004:** Comparison results of varying *K* values.

*K*	Uncompression (Bits)	Compression (Bits)	DRS (%)
	$\left(G_{1},G_{2}\right)$	$\left(G_{1},G_{2}\right)$	$\left(G_{1},G_{2},G\right)$
0	256, 240	91, 81	64.45, 66.25, 65.32
1	256, 240	94, 88	63.28, 63.33, 63.31
2	256, 240	88, 88	65.63, 63.33, 64.52
3	256, 240	86, 88	66.41, 63.33, 64.92
4	256, 240	86, 76	66.41, 68.33, 67.34
5	256, 240	79, 79	69.14, 67.08, 68.15

**Table 5 sensors-19-03190-t005:** Comparison between compression and transmission steps for datasets with different size.

Dataset	Algorithm	Compression Step				Transmission Step		
		#Bits	Time (s)	DRS (%)	Energy (mJ)	Time (s)	#Packets	Energy (mJ)
sine-like (352 bits)	RLE	464	0.023	−31.82	0.275	0.015	0.453	0.781
	*K*-RLE	320	0.023	19.09	0.303	0.011	0.313	0.473
	LEC	149	0.037	57.67	0.313	0.005	0.146	0.222
	LZW	207	0.110	41.19	0.396	0.008	0.202	0.309
	D-RLE	132	0.075	62.50	0.327	0.004	0.129	0.203
chaotic (1176 bits)	RLE	1504	0.076	−27.89	0.919	0.050	1.469	2.530
	*K*-RLE	1168	0.077	0.68	1.014	0.041	1.141	1.728
	LEC	733	0.122	37.67	1.047	0.024	0.716	1.092
	LZW	648	0.367	44.90	1.323	0.025	0.633	0.967
	D-RLE	616	0.249	47.62	1.092	0.019	0.602	0.947
simulated temperature (1600 bits)	RLE	1600	0.103	10.00	1.250	0.053	1.563	2.692
	*K*-RLE	1280	0.104	20.00	1.379	0.045	1.250	1.893
	LEC	651	0.166	59.31	1.424	0.021	0.636	0.970
	LZW	1584	0.500	11.00	1.800	0.061	1.547	2.363
	D-RLE	416	0.339	74.00	1.486	0.013	0.406	0.640
temperatureHr (768 bits)	RLE	752	0.049	12.08	0.600	0.025	0.734	1.265
	*K*-RLE	512	0.050	33.33	0.662	0.018	0.500	0.757
	LEC	528	0.080	31.25	0.684	0.017	0.516	0.787
	LZW	504	0.240	34.38	0.864	0.020	0.492	0.752
	D-RLE	204	0.163	73.44	0.713	0.006	0.199	0.314
temperatureMin (46,080 bits)	RLE	984	2.963	97.87	36.005	0.033	0.961	1.655
	*K*-RLE	656	3.009	98.58	39.713	0.023	0.641	0.970
	LEC	6240	4.780	86.46	41.018	0.202	6.094	9.299
	LZW	4230	14.396	90.82	51.847	0.164	4.131	6.310
	D-RLE	467	9.776	98.99	42.802	0.015	0.456	0.718

**Table 6 sensors-19-03190-t006:** Total energy use for the datasets with different sizes.

Dataset	Total Energy Use (mJ)				
	RLE	*K*-RLE	LEC	LZW	D-RLE
sine-like	1.056	0.777	0.535	0.705	0.530
chaotic	3.449	2.741	2.139	2.290	2.040
simulated temperature	3.942	3.272	2.394	4.163	2.126
temperatureHr	1.865	1.419	1.470	1.616	1.027
temperatureMin	37.660	40.683	50.317	58.157	43.520

**Table 7 sensors-19-03190-t007:** Comparison between compression and transmission steps for datasets with the same size of 46,080 bits.

Dataset	Algorithm	Compression Step				Transmission Step		
		#Bits	Time (s)	DRS (%)	Energy (mJ)	Time (s)	#Packets	Energy (mJ)
sine-like	RLE	20,352	3.044	55.83	34.903	0.680	19.875	34.239
	*K*-RLE	16,056	3.060	65.16	37.493	0.570	15.680	23.749
	LEC	6876	4.862	85.08	40.778	0.223	6.715	10.246
	LZW	6588	14.156	85.70	50.407	0.256	6.434	9.828
	D-RLE	4476	10.087	90.29	42.703	0.140	4.371	6.883
chaotic	RLE	21,888	2.991	52.50	34.651	0.731	21.375	36.823
	*K*-RLE	18,816	3.126	59.17	35.928	0.668	18.375	27.832
	LEC	8796	4.762	80.91	40.538	0.285	8.590	13.107
	LZW	7776	14.052	83.13	49.407	0.302	7.594	11.600
	D-RLE	7332	9.367	84.09	41.047	0.230	7.160	11.275
simulated temperature	RLE	19,200	3.023	58.33	34.918	0.641	18.750	32.301
	*K*-RLE	16,560	3.005	64.06	36.986	0.588	16.172	24.495
	LEC	7812	4.836	83.05	40.596	0.253	7.629	11.641
	LZW	19,008	13.346	58.75	49.447	0.737	18.563	28.355
	D-RLE	7332	9.967	84.09	41.630	0.230	7.160	11.275
temperatureHr	RLE	45,120	2.960	2.08	34.577	1.506	44.063	75.906
	*K*-RLE	30,720	3.006	33.33	35.976	1.091	30.000	45.439
	LEC	31,680	4.755	31.25	39.098	1.028	30.938	47.208
	LZW	30,240	13.368	34.38	47.657	1.173	29.531	45.111
	D-RLE	12,240	9.420	73.44	41.023	0.384	11.953	18.823
temperatureMin	RLE	984	2.963	97.87	36.005	0.033	0.961	1.655
	*K*-RLE	656	3.009	98.58	39.713	0.023	0.641	0.970
	LEC	6240	4.780	86.46	41.018	0.202	6.094	9.299
	LZW	4230	14.396	90.82	51.847	0.164	4.131	6.310
	D-RLE	467	9.776	98.99	42.802	0.015	0.456	0.718

**Table 8 sensors-19-03190-t008:** Total energy use for the datasets with the same size of 46,080 bits.

Dataset	Total Energy Use (mJ)				
	RLE	*K*-RLE	LEC	LZW	D-RLE
sine-like	69.142	61.242	51.025	60.235	49.586
chaotic	71.474	63.760	53.646	61.007	52.322
simulated temperature	67.218	61.481	52.237	77.803	52.906
temperatureHr	110.483	81.415	86.306	92.768	59.846
temperatureMin	37.660	40.683	50.317	58.157	43.520

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
