# Peer review of "Differential Run-Length Encryption in Sensor Networks"

_sensors, 2019, doi:10.3390/s19143190_

Round 1

Reviewer 1 Report

I think it is an interesting paper for the remote sensor community and the IoT community. The results stand on their own. The paper has well presented metrics and the psuedocode makes the logic of the paper easy to follow. I wonder if the overall impact of savings on transmission power will be diluted when factoring in protocols involving sensor sleep-wake scheduling or anycasting.

Figure 5 radar chart is difficult to read at 100% magnification due to the small text. Also, it would help if some of the text describing the contents of each radar chart is moved into the caption so that the reader does not have to flip back and forth between the figure and the text.

Author Response

Point 1: I think it is an interesting paper for the remote sensor community and the IoT community. The results stand on their own. The paper has well presented metrics and the psuedocode makes the logic of the paper easy to follow. I wonder if the overall impact of savings on transmission power will be diluted when factoring in protocols involving sensor sleep-wake scheduling or anycasting.

Figure 5 radar chart is difficult to read at 100% magnification due to the small text. Also, it would help if some of the text describing the contents of each radar chart is moved into the caption so that the reader does not have to flip back and forth between the figure and the text.

The font sizes in Fig.5 have been increased as shown in the attached file.

Reviewer 2 Report

Good manuscript.

I would suggest to explore in future works how to impact on the only dataset (temperatureMin) which the proposed technique looks not efficient for. 

Author Response

Point 1: I would suggest to explore in future works how to impact on the only dataset (temperatureMin) which the proposed technique looks not efficient for. 

Thanks for your suggestion. According to the temperatureMin dataset in Fig.5(e), compared to RLE and K-RLE, D-RLE takes the longest compression time, while D-RLE’s compression time is moderate when compared to LEC and LZW. Time efficiency can be improved in future works by combining data fusion with our proposed technique. In a cluster, the cluster head may collect all data sensed by neighboring nodes, compress them and transmit the fused data to the base station. A neighboring node does not compress data. It simply sends raw data to its cluster head.

Reviewer 3 Report

The paper presents a new compression algorithm specifically designed to save energy consumption for wireless sensor networks. This is a area with a long history and a lot of literature work can be found for this research topic. The proposed algorithm seems feasible, although I didn't check thoroughly. Extensive evaluation was carried out the verify the effectiveness of the proposed algorithm against four other baseline algorithms on five different data sets. Overall, I enjoy reading this paper which is well-written: clear and easy to follow. 

I suggest the authors consider improving this paper from the following perspectives.

(1) In introduction, please give more quantitative background on this research topic. For example, energy is of course an important criterion for wireless sensor networks (WSNs). But how bad is it, quantitatively? In other words, having a compressed data set with 10% in size implies 90% in energy consumption? The information might be available in the evaluation section, but the readers are more interested in knowing this upfront.

(2) The introduction should be elaborated a bit. At the very least, the introduction should highlight the research challenge. For instance, why is saving energy consumption a hard problem, and how does the proposed algorithm, at a very high level, novelly solve this problem? What's the intuition behind the algorithm? (or, how did you come up this algorithm?)

(3) The authors propose a set of new algorithms for the compression. And yet, there is no correctness proof or complexity analysis for any of them. It is understandable to skip the correctness or even complexity for trivial algorithms, but I suggest authors to carefully go through the proposed algorithms and reconsider whether to include the correctness proof and complexity analysis.

(4) The evaluation is not clear about which test bed is used. For instance, how many sensor nodes are deployed? What's the configuration of the sensor nodes? Or are all experiments carried out by simulation? Please make it clear at the beginning of the evaluation section. 

(5) Please consider add more recent references in this area (i.e., energy consumption in WSN). For instance, I know this journal (Sensors) and other leading conferences recently published a few papers in this field:

* https://www.mdpi.com/1424-8220/19/5/1002

* https://www.mdpi.com/1424-8220/19/4/784

* https://dl.acm.org/citation.cfm?doid=3225058.3225073

(6) Some language issues. For instance, Line 204, "It is depended" -> "It depends", "quality lost" -> "quality loss", etc. There must be more issues as I didn't check language thoroughly. Please carefully proofread the paper in the revised submission.

Author Response

Point 1: In introduction, please give more quantitative background on this research topic. For example, energy is of course an important criterion for wireless sensor networks (WSNs). But how bad is it, quantitatively? In other words, having a compressed data set with 10% in size implies 90% in energy consumption? The information might be available in the evaluation section, but the readers are more interested in knowing this upfront.

Point 2: The introduction should be elaborated a bit. At the very least, the introduction should highlight the research challenge. For instance, why is saving energy consumption a hard problem, and how does the proposed algorithm, at a very high level, novelly solve this problem? What's the intuition behind the algorithm? (or, how did you come up this algorithm?)

As suggestion In highlight the research challenge, we mentioned that energy is a hard problem that there are many solutions

Based on point 1 and 2, the introduction is improved as shown between line 14-30 in the attached file.

Point 3: The authors propose a set of new algorithms for the compression. And yet, there is no correctness proof or complexity analysis for any of them. It is understandable to skip the correctness or even complexity for trivial algorithms, but I suggest authors to carefully go through the proposed algorithms and reconsider whether to include the correctness proof and complexity analysis.

Pseudocodes 1 to 3 have the linear time complexity O(n) and perform best in their own characteristics, not for general datasets. This gap stimulates how we can combine each strong point of those algorithms to compress the data. To this end we have developed Differential Run Length Encryption (D-RLE), which also has the linear time complexity O(n), and will explain its concept in the paper in Pseudocodes 4-6.

Point 4: The evaluation is not clear about which test bed is used. For instance, how many sensor nodes are deployed? What's the configuration of the sensor nodes? Or are all experiments carried out by simulation? Please make it clear at the beginning of the evaluation section. 

In our testbed, we assumed that data sets are already stored in a sensor node. The sensor node could be a cluster head that collects data either from itself or from neighboring nodes. Later the sensor node performs data compression and wirelessly send compressed data to a receiver.

Point 5: Please consider add more recent references in this area (i.e., energy consumption in WSN). For instance, I know this journal (Sensors) and other leading conferences recently published a few papers in this field:

* https://www.mdpi.com/1424-8220/19/5/1002

* https://www.mdpi.com/1424-8220/19/4/784

* https://dl.acm.org/citation.cfm?doid=3225058.3225073

We referred three suggested papers in the reference No. 5, 7 and 14 as follows.

[5] Wang, J.; Gao, Y.; Liu, W.; Sangaiah, A.K.; Kim, H.J. Energy Efficient Routing Algorithm with Mobile Sink Support for Wireless Sensor Networks. Sensors 2019, 19.

[7] Cheng, J.; Gao, Y.; Zhang, N.; Yang, H. An Energy-Efficient Two-Stage Cooperative Routing Scheme in Wireless Multi-Hop Networks. Sensors 2019, 19.

[14] Wang, J.; Al-Mamun, A.; Li, T.; Jiang, L.; Zhao, D. Toward Performant and Energy-efficient Queries in Three-tier Wireless Sensor Networks. Proceedings of the 47th International Conference on Parallel Processing; ACM: New York, NY, USA, 2018; ICPP 2018, pp. 42:1–42:10.

(6) Some language issues. For instance, Line 204, "It is depended" -> "It depends", "quality lost" -> "quality loss", etc. There must be more issues as I didn't check language thoroughly. Please carefully proofread the paper in the revised submission.

In the attach file, we have improved the language issues from “it is depended” to “It depends” and from "quality lost" to "quality loss" in line 218.

Round 2

Reviewer 3 Report

All my comments have been addressed; I thus recommend an acceptance of this paper.